**METHOD**

# MCProj: metacell projection for interpretable and quantitative use of transcriptional atlases

Oren Ben-Kiki[1], Akhiad Bercovich[1], Aviezer Lifshitz[1], Ofir Raz[1], Dror Brook[1] and Amos Tanay[1*]

*Correspondence:
amos.tanay@weizmann.ac.il

[1] Department of Applied
Math and Computer Science,
Weizmann Institute of Science,
Rehovot, Israel

## Abstract

We describe MCProj—an algorithm for analyzing query scRNA-seq data by projections over reference single-cell atlases. We represent the reference as a manifold of annotated metacell gene expression distributions. We then interpret query metacells as mixtures of atlas distributions while correcting for technology-specific gene biases. This approach distinguishes and tags query cells that are consistent with atlas states from unobserved (novel or artifactual) behaviors. It also identifies expression differences observed in successfully mapped query states. We showcase MCProj functionality by projecting scRNA-seq data on a blood cell atlas, deriving precise, quantitative, and interpretable results across technologies and datasets.

## Background

Cell atlases are being charted in numerous domains and applications, harnessing the power of single cell genomics to an increasing array of supporting technologies and algorithms. Constructing a cell atlas typically involves sample collection, cell profiling, normalization and noise removal, and inference of cell type or cluster structure. This ultimately leads to extensive annotation and comparison with prior literature that link cell maps with function, developmental contexts, and proposed dynamics. Atlases are then hoped to become a standardized reference for a tissue or a process, and it is of utmost importance to be able to use them quantitatively and systematically when analyzing new data, taking true advantage of the efforts put into their construction. Most importantly, atlases are not collections of profiled cells, but instead represent distributions of molecular states inferred from such cells, in anticipation of facilitating the mapping and "positioning" (here defined as "projection") of newly collected, currently still unobserved cells.

Current tools in the field are facilitating atlas construction and integration of new data into larger existing atlases [1–9] bridging technical artifacts such as batch effects and technology differences. But integration of multiple datasets may not be the method of choice when aiming at the analysis of a new dataset vis-à-vis an existing reference atlas.

An approach distinguishing the reference atlas from the new ("query") data [10] can help create a common ground for analysis of new experiments, in particular experiments involving perturbation and stimulation of cells or tissues, or those involving comparison of atlases arising from individuals varying in their genetic background, disease state or environmental exposure. Approaches for mapping cells on reference data may use a variety of strategies [3, 11, 12] including transfer learning [4, 13, 14] or simple correlation matching of cells to atlas distributions [15].

We suggest that gene expression and its quantification by scRNA-seq data is sufficiently quantitative, robust, and universal, such that new data (representing new sampled cells from similar tissues) should be interpretable using the precise quantitative RNA-seq distributions as defined by the reference atlas. Furthermore, we suggest that any quantitative discrepancy between the new cells and the reference is of potential interest and should be tagged and further classified for downstream analysis by the user. Opaque integration or projection strategies can be used, but these should not substitute the direct test for a quantitative match between RNA distributions. If and when a quantitative projection is achieved successfully, the reference atlas can become a true anchor for studying how cell states and dynamics are modulated or perturbed between experiments. To facilitate this strategy, we developed MCProj, an algorithm for quantitative analysis of query scRNA-seq given a reference atlas. The algorithm transforms single cells to quantitative states using a metacell representation of the atlas and the query. It then infers each query state as a mixture of atlas states and tags cases in which such inference is imprecise, suggestive of novel or noisy states in the query. We describe below how this approach can be implemented efficiently, and how gene expression distributions can be corrected in cases of technology differences, such that quantitative projection is still effective. We also describe strategies for visualizing projection results interactively.

## Results

### Overview of metacell based atlases and the MCProj algorithm

We represent a reference atlas as an annotated metacell model [16, 17]. This representation includes a *set of states* that are defined by a quantitative gene expression distribution and an annotation term. Importantly, the atlas representation does not include the single cell profiles used for generating the atlas, as it aims at representing canonical cell states that are likely to be recurrently observed when assaying a tissue or process of interest. States can be linked together into a manifold that we represent as a graph structure, and the graph can be embedded into a 2D space for visualization. Such embedding is however not truly a part of the reference model and is meant only for ease of interpretation or as a basis for further modeling. The annotation terms that are used are also intended for simplification and streamlining downstream analysis. But as most cellular state spaces involve a continuum of states that cannot unambiguously be demarcated into cell types, the native resolution of the atlas is the metacell states and not the annotated types. Additionally, the atlas definition includes a set of genes that are selected as *features*, and a list of lateral genes that should be forbidden from use as features as likely representative of transient behaviors like the cell cycle, but are otherwise part of the atlas state definitions. The atlas also includes a list of noisy genes that are exempted from

perfect quantitative matching due to known unstable or bursty expression distribution. Finally, the atlas specifies, if possible, a small number of genes that are essential markers for each cell type annotation term. The assumption is that cells from each type should match quantitatively (but not necessarily perfectly) the expression profile of the feature genes of a metacell from the type, with special attention to the matching of the essential genes. We note that some genes are excluded from any analysis as representative of processes contributing substantially to bias that may lead to errors in estimating transcript frequencies (e.g., mitochondrial transcripts), but as these are technically not part of the atlas definition, we can simply ignore them.

To project a query scRNA-seq dataset onto a given reference atlas, we first infer a metacell structure over it. The metacell size parameters (target number of UMIs per metacell) can be decreased when the query is a relatively small dataset, in order to generate more metacells (even if these would be less precise) and maximize resolution on the query. Given a set of query metacells, the algorithm aims at representing these as a weighted average of atlas states. This is done while iteratively identifying genes that cannot be used for projection, and while inferring correction terms for genes that can be used for comparison but are recovered and sequenced with different efficiency between the query and atlas. In any case, all comparisons of the query and atlas are quantitative—avoiding the use of K-nn graphs or other non-parametric embeddings and searching directly for a projection that explains the query gene expression states using atlas gene expression states. The algorithm can then tag query states that could not be explained (since they are noisy or novel) and highlights genes that are differentially expressed in the query states compared to the atlas.

### MCProj robustly matches existing atlas states

To test MCProj systematically, we constructed an atlas representing cell states in the human bone marrow (BM). We used HCA [18] data on 378,000 cells to infer 2684 metacells. The atlas includes data for 27,261 genes, out of which 3322 were identified as features for projections. 5739 genes were tagged as lateral (not affecting projection). This atlas specified no noisy genes. Its derivation followed the exclusion of 6417 poorly annotated, non-coding and/or mitochondrial genes from the raw count matrix (see "Methods" for definitions). Atlas MCs were filtered and annotated into 19 cell types using prior knowledge (Fig. 1A–C). We defined 1–4 essential marker genes for each type. We first wished to study MCProj robustness when projecting cells from states that are well represented in the atlas. To test this, we implemented a cross-validation-like design. We considered a *master* BM atlas model using data from 8 batches representing 8 donors. For each batch, we removed its cells from the atlas and computed a *punctuated* atlas based only on the remaining cells. We then treated cells from the removed batch as the query dataset and used MCProj to project it onto the punctuated atlas. Using the projection, we found that 99.85% of the left-out metacells were mapped onto the atlas such that they were similar to their projected image in the atlas (see the "Methods" section for details). MCProj was using 68–77% of the feature genes for fitting the query. Furthermore, the majority of mapped cells were linked with a cell type that was consistent with their original master-atlas annotation (Fig. 2A). When apparent annotation mismatches were observed, they involved shifting of related cell types over differentiation gradients,

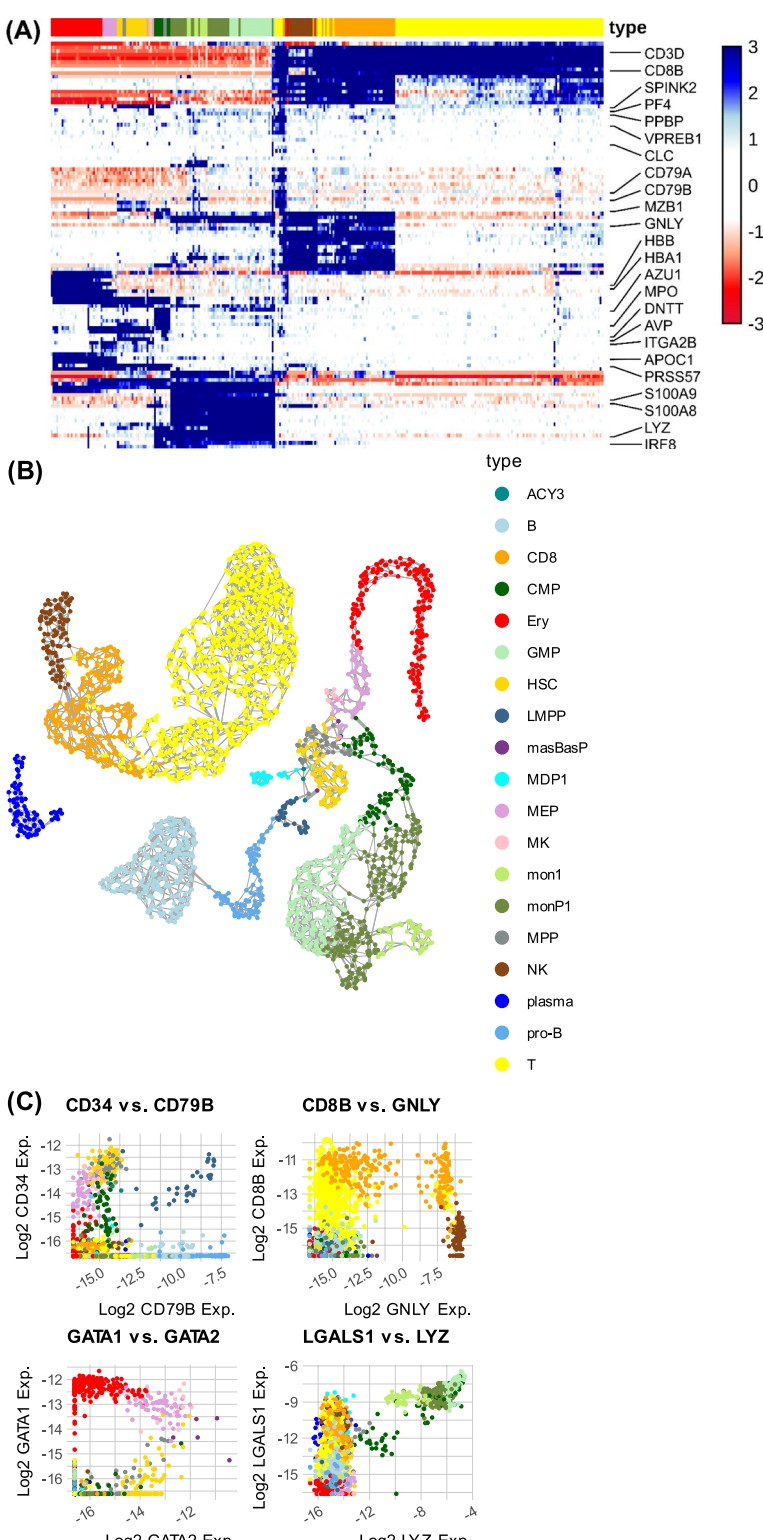

**Fig. 1** Human cell bone marrow atlas. This atlas was computed using the Metacells-2 algorithm and was used as a basis for the analysis below. **A** Heatmap of feature gene expression in the atlas metacells. Annotated cell types are color coded at the top. **B** 2D UMAP projection of the atlas metacells. Edges are visualizing a metacell similarity graph. **C** Quantitative gene expression comparison over metacells, shown for select genes that were defined as essential for a cell type annotation

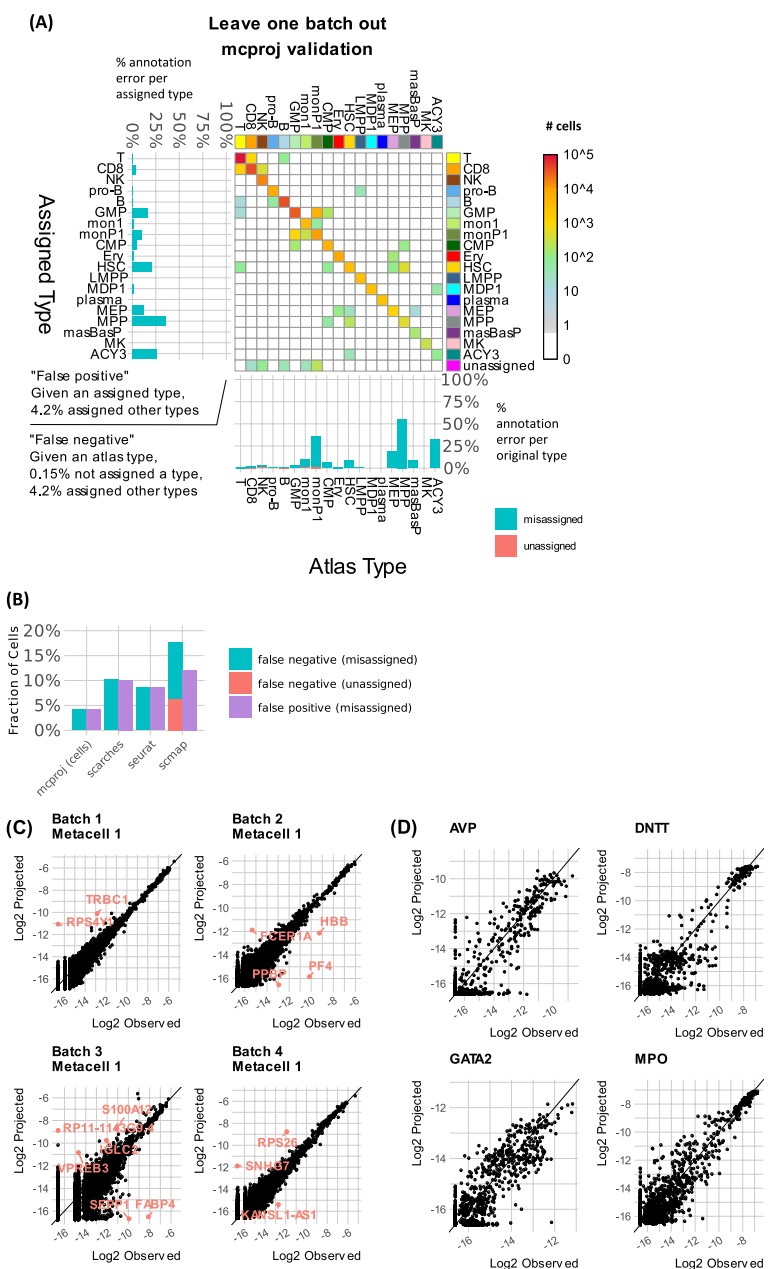

**Fig. 2** Validation: leave-one-batch-out projections. Data from a series of tests projecting a query consisting of cells from one batch, onto an atlas computed from cells from all other batches. For assessing robustness, we use the labels for each cells as inferred from analysis of the atlas including all batches as shown in Fig. 1 above. **A** Counting cells classified according to their reference annotated type and the type assigned to their query metacell by MCProj. Summary of false-positive and false-negative associations are shown as bar-plots (left—fraction of misclassified query cells per MCProj assigned type, bottom—fraction of misclassified query cells per original type). **B** The fraction of query cells which were not successfully projected and which were assigned a type different than the expected type, for each of the methods we compared. **C** Example of query metacells observed vs. projected expression over all genes. **D** For some example genes, shown are observed vs. projected expression levels over all query metacells (overlaying data from all batches)

such as those defining HSC differentiation to MPP or MEP, or those associated naïve T-cells with CD8+ T-cells. This analysis confirmed the ability of MCProj to match quantitatively cells onto the atlas, when the atlas indeed represents their states.

We compared MCProj performance to scMAP [11], scARCHES [13] (using SCANVI [4]), and Seurat [3] data transfer published single cell projection schemes, using a similar cross-validation strategy (Fig. 2B, Additional file 1: Fig S1-S3). MCProj failed to assign a label to 0.15% of the cells and misassigned (assigned a different label) 4% of the cells. In contrast, scMAP showed a higher rate of unassignment (6%) and misassignment (12%) (Additional file 1: Fig S3). For scARCHES and Seurat, we picked a score threshold to match the unassignment rate of mcProj (0.15%), which yielded higher rates of misassigned labels (10% for scARCHES and 9% for Seurat, Additional file 1: Fig S1-S2). We note that using the transfer learning strategy could result in losing key progenitor states (MEP, MPP) altogether. These analyses suggest that MCProj strategy of first forming metacells on the query (including removal of outlier cells), and then matching quantitatively the query states over the atlas state can be more accurate, at least in scenarios in which the query perfectly matches atlas states.

An important feature of MCProj is the analysis of quantitative gene expression distribution in the query and atlas state, which is intended to highlight differential expression even in matched states. In our cross-validation, such differential expression is expected to be non-significant. Indeed, 68–78% of the atlas genes were fitted by the algorithm that reconstructed query metacell gene expression with high precision (exemplified in Fig. 2C for four query MCs). Only a few problematic and batch-effect-prone genes showed a strong bias (for example, the Y encoded RPS4Y1 gene in MC#1 of batch 1, or the IGK/IGH genes in MC#1 of batch 4). Overall, cross-validation reflects a good reconstruction of the expression distribution of key genes in the query using a mixture of atlas states (Fig. 2D), where the remaining variance represents mainly the uncertainty in the inference of gene expression frequency within each metacell. We note that this type of quantitative comparison is generally missing from existing atlas projection tools, as these are designed to match cells rather than gene expression distributions.

### MCProj tags novel query states and compares them to atlas states

To test whether MCProj also specifically tags novel transcriptional states in the query if such states exist, we have considered the 19 metacell types included in the bone marrow atlas. We removed all metacells associated with each type in turn from the atlas to create a punctuated atlas containing only states representing the remaining 18 cell types. We then used MCProj to project the extracted cell-type metacells onto the punctuated atlas. Finally, we compared for each extracted cell the type assigned to it by MCProj to its original type in the atlas (Fig. 3A). The overall results demonstrate that MCProj successfully tags metacells representing novel states. Ten left-out types queries were tagged as globally unassigned given low percentage of matching query feature gene ranges (0–26%). Nine additional types showed sufficient correlation with an existing atlas state to define some assignments. But such assignment used lower fractions (45–71%) of the query feature genes and showed low quantitative $R^2$ values of the fitted (projected) genes between the atlas and the query (Fig. 3B).

When a left-out type was matched to the atlas, this typically indicated highly related types, such as annotation of left-out CD8+ T cells as T-cells. In cases where matching of a left-out type was inferred by MCProj, the quantitative gene expression comparison (Fig. 3C) of query and matched atlas states were shown to be instrumental for

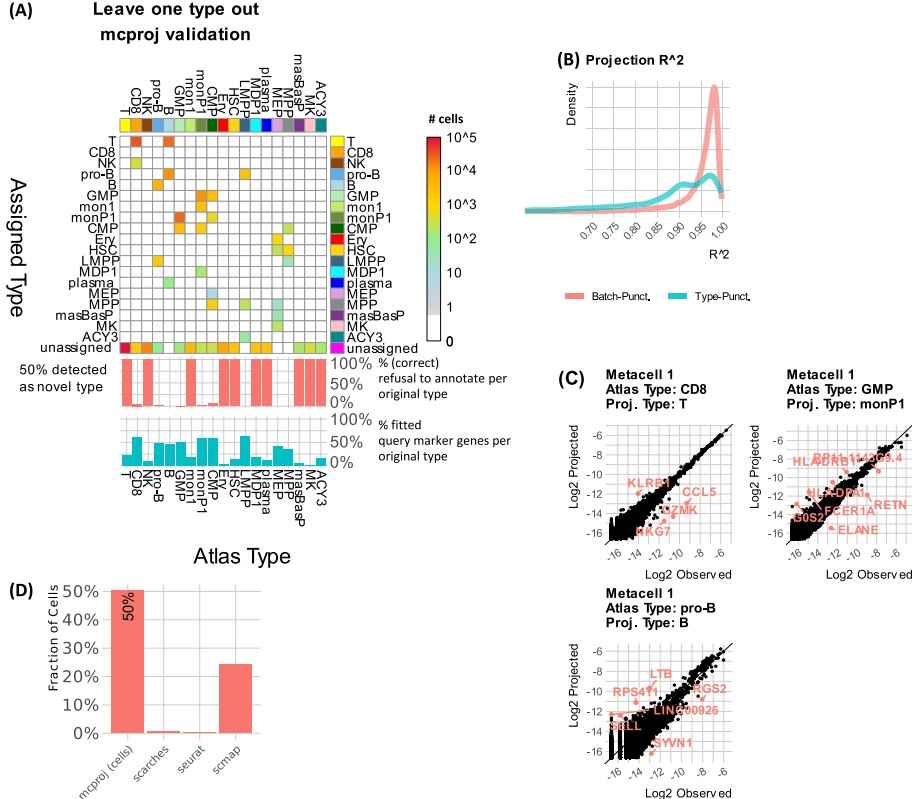

**Fig. 3** Validation: leave-one-type-out projections. Data from tests projecting queries including all cells annotated with one type onto an atlas from which all cells of this type were removed. **A** Counting cells classified according to their reference annotated type and the type assigned to their query metacell by MCProj. Middle (red) bars summarize the fraction of query cells with any assigned types (note that assigned types may still be associated with differential projected/atlas expression). Lower (blue) bars show the fraction of query feature genes that could be considered for fitting by MCProj (with all other genes excluded from fitting given mismatching gene expression range, see "Methods"). **B** Distribution $R^2$ values across all metacells between the actual gene expression and its projection on the batch- and type-punctuated atlases. **C** Examples of metacells that were mapped to a punctuated atlas, showing their observed vs. projected gene expression and highlighting key differential genes. **D** The fraction of query cells which were not successfully projected (indicating a novel type) for each of the methods we compared—larger values represent a better outcome

understanding the query differential expression relative to the atlas. For example, GZMK and CCL5 are higher in specific projected Cd8+ T-cells compared to their matched atlas naïve T-cells, indicating the initiation of a differentiation process. Similarly observed are expression differences of LTB and SELL (marking differentiation of pro-B cells toward B-cells), or expression differences of more ELANE and less FCER1A and MHC-II genes when projecting GMP cells and matching them to monocyte progenitors.

Analysis of the performance of scMAP, scARCHES, and Seurat (using the same score threshold as above) when re-projecting types on punctuated atlases showed lower capacity to tag novel states compared to MCProj (Fig. 3D, Additional file 1: Fig S4-S6). scMAP unassigned (detected as novel) a lower fraction of 24% of the cells, while the other methods detected significantly lower rates of novel states (0.59% for scARCHES and 0.34% for Seurat). In some cases, the integration and harmonization approaches resulted in rather drastic classification errors (e.g., B cells and CD8+ T-cells). These data indicated that

comparison of quantitative gene expression distribution rather than single cells endows MCProj with enhanced specificity for detecting and characterizing novel transcriptional states in query data.

### MCProj corrects for technology-linked systematic biases

MCProj compensates for systematic biases introduced by different scRNA-seq technologies by inferring (if needed) for each gene a multiplicative correction factor that balances the expression trend in the query to the one in the atlas. MCProj also excludes genes with poor quantitative (up to a factor) query-atlas match. The rationale behind MCProj corrective approach is that any technology or platform bias that affects transcripts capture efficiency independently over molecules and due to the RNA sequence itself must induce a multiplicative effect on gene expression frequency that remains homogeneous over the entire manifold. Such effects can therefore be inferred directly as a single interpretable parameter per gene. We note that this mechanism will intentionally not compensate for other more complex forms of systematic bias. For example, MCProj will not correct non-multiplicative or cell-type specific effects such as those induced by regulation of nuclear export and generating differences between scRNA-seq and snRNA-seq. Instead, MCProj may mask genes affected non-multiplicatively from the set used in fitting query to atlas expression. Restricting the correction to a conservative global multiplicative correction factor per gene ensures that the process will not mask important biological differences between the two data sets.

We tested the robustness of this strategy by projecting data characterizing BM CD34+ from 8 donors that was generated using 10x-v3 technology [19] over the BM atlas that was generated using 10x-v2 technology. For each donor, we computed metacells from the donor's cells and used MCProj to project them on the atlas, with and without allowing for automatic correction of systematic technology biases. MCProj was using 43–78% of the feature genes for fitting the query. Figure 4A shows the atlas metacells that were used by MCProj to represent the donors' metacells. Analysis of $R^2$ values between query and atlas projections (Fig. 4B) showed MCProj can use the inferred correction factors for automatic correction of systematic technology biases to reach a similar projection quality as for the batch-punctuated experiment above. Figure 4C shows several examples of genes for which the two protocol versions differ in a way that is captured by a single corrective multiplicative factor, and Fig. 4D reports on the query-atlas correlation of all genes compared to their inferred multiplicative correction, marking the set of genes used by MCProj to apply its parametric gene correction.

To test the robustness of the automatic correction procedure, we went back to the cross-validation design leaving out one donor at a time from the BM, 10x-v2-based atlas. To emulate technology-linked biases, we selected genes for manipulation prior to projection of the query (using the same genes corrected in the real data). We synthesized bias with increasing factors (Fig. 4E) or increasing number of genes (Fig. 4F) while recording the query to atlas quantitative $R^2$ values, and the overall rate of query to atlas metacell matching. This strongly supported the robustness of the correction mechanism showing genes are automatically compensated for by the algorithm, with minimal impact on the overall result. We note that applying the correction mechanism results in some overfitting but this remains minimal, as shown by the ~1% improved $R^2$ in our tests.

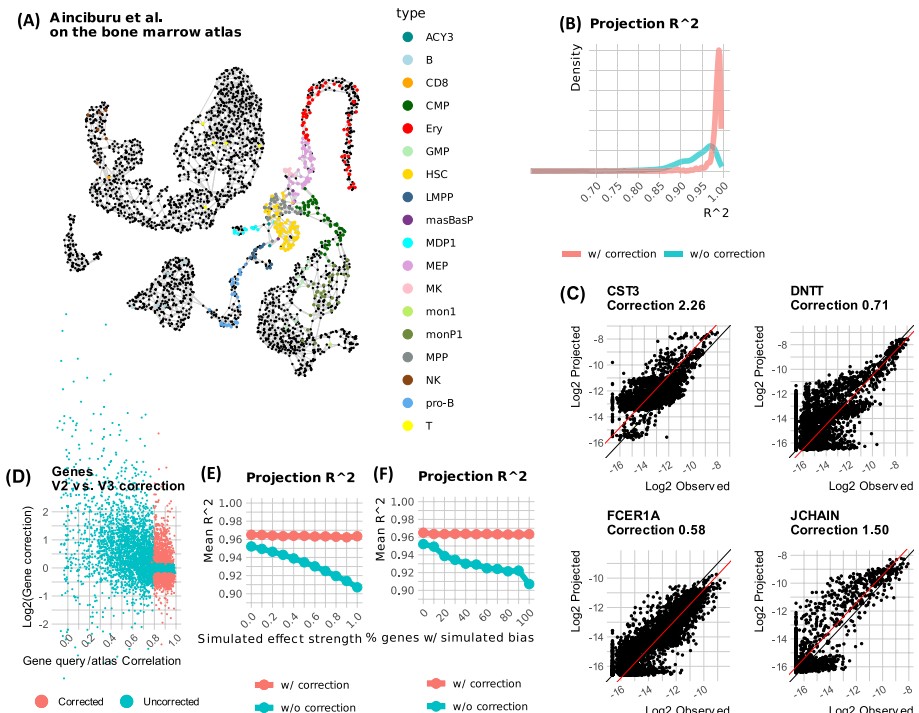

**Fig. 4** Automatic technology corrections. **A** Showing atlas metacells used to represent query metacells when projecting 10x-v3 blood data on the 10x-v2 BM atlas (as shown in Fig. 1). **B** Shown are distributions of observed vs, projected expression $R^2$, with and without automatic gene correction. **C** Examples of corrected genes. Correction factor (constant in log-scale) is shown as a red line. **D** MCProj applies auto-gene correction when the correlation between observed and projected expression overall all metacell is high, and the correction factor is of a large magnitude. Shown here are correlation and optimal correction factor values for all genes when applying MCProj as in **A**. Red dots represent genes that were selected for correction. **E** $R^2$ values of metacell genes successfully projected on the batch-punctuated atlas (Fig. 2), when varying the strength of a synthetic technology bias applied to the genes of the batch, where the full strength is identical to the one computed in Fig. 4C. **F** Similar to **E**, but when varying the fraction of genes that are synthetically modified by a multiplicative factor

We note that MCProj, as well as other projection algorithms, is not significantly changing its type annotations when introducing multiplicative biases to genes (Additional file 1: Fig S7). The corrections performed in MCProj are important for facilitating quantitative differential gene expression analysis of the query and the atlas, going beyond type annotation per se.

## Discussion

We have introduced MCProj—a new algorithm for projecting single-cell RNA-seq data sets on annotated single-cell atlases. The design of MCProj leverages the advantages of using a metacell model for both the atlas and query data sets. Metacell profiles provide robust estimates for gene expression for specific cell states, as opposed to sparse cell profiles which must be grouped together to provide such estimates, or cell type clusters which encompass variations within each coarse-grained cell type label. This allows MCProj to provide a more precise projection compared to cell-based approaches, modeling each query metacell as a weighted average of a small number of atlas metacells. Given a query dataset, MCProj results include cell type assignment

and a quantitative projected profile for each mapped query metacell, together with the detection of metacells representing gene expression states that are inferred not to be present in the atlas. Even for metacells that are mapped onto the atlas, MCProj provides detailed differential gene expression between the query and the reference atlas transcriptional state. MCProj is direct and interpretable and is also computationally efficient, performing well without requiring strong CPU or GPU clusters.

This quantitative projection and the provided built-in quality controls allow the analyst to focus on the similarities and differences between the query and the atlas, to identify cell behaviors common to both as well as detect novel cell behaviors unique to the query data set. We show this can provide much better specificity and sensitivity compared to approaches working directly with sparse single-cell profiles. Furthermore, we suggest that as atlases become more comprehensive and better annotated, the utility of the quantitative approach will further increase, as would the importance of converging multiple single-cell RNA-seq datasets into one common quantitative set of gene expression profiles describing a tissue or process of interest.

When seeking the ultimate convergence of reference atlases, it is important to understand and control for systematic biases induced by different technologies for capturing RNA from single cells. MCProj is natively correcting for biases that are working independently on single molecules (during, e.g., reverse transcription, tagging, amplification, mapping)—since these must result in a single multiplicative factor per gene and technology, representing the (relative) average preference for capturing and sequencing mRNA from each gene. Biases that modulate the estimation of RNA concentrations for groups of genes can create cell-type or cell-state-specific biases that cannot be fully corrected by MCProj. Moreover, some acquisition technologies may profile nuclear RNA or employ other fractionation strategies, resulting in technology biases that can represent de-facto gene regulation (e.g., nuclear export). Such regulatory biases should in principle *not* be "corrected" by a projection algorithm, although a projection algorithm may assist in characterizing them. MCProj can de-facto allow projection of snRAN-seq on scRNA-seq data but this will force a large fraction of the genes (and in particular those showing differential nuclear/cytoplasm concentrations) to be excluded from fitting. Future atlases can be enhanced to define for each state (or metacell) a composite distribution that will model explicitly nuclear and cytoplasmatic gene expression. Ideally, in such atlases, it will be desirable to calibrate gene expression estimation in at least one technology using an independent counting technique (such as RNA-FISH). Calibrated atlases will offer an absolute scale for gene expression, allowing MCProj corrections to be interpreted over a fixed scale rather than relative to one, arbitrary, atlas technology.

MCProj projection of a query over an expression atlas produces rich data that is best explored interactively. To facilitate such exploration, we extended our metacell viewer tools with functionality for globally exploring query projection over manifold 2D projection for in-depth analysis of query vs. atlas metacell differential expression, and for gene-centric analysis evaluating differential gene expression for specific genes over the entire query. These views allow users to first annotate their query data with matching atlas cell types/states, detect novel types by in-depth analysis of expression states, and examine potential differential expression in query metacells that are

matched with the reference, in order to detect query-specific effects (e.g., patient-specific expression, treatment-effects, mutant effects).

Further work must be channeled toward the ability to incrementally update and extend annotated metacell atlases given additional data sets to cover new cell states and better sample existing ones. The aim can be to minimize the supervised effort of repeated re-annotation of extended atlases. An additional challenge ahead is to formulate multi-layered atlases that incorporate epigenetics or proteomics data, and devise effective query functionality over them.

## Conclusions

MCProj provides a quantitative interpretation of scRNA-seq data given a reference atlas. As large atlases evolve to provide better coverage of key transcriptional manifolds, MCProj can facilitate analysis of new datasets following perturbation, stimulation, or patient sampling over a stable and highly interpretable common reference.

## Methods

### The atlas metacell model

A metacell atlas is built using gene sets, metacell expression distributions, metacell cell types, and type annotations as follows:

A. *Gene sets* are defined over a set of genes that are characterized by one or more names and identifiers. It is assumed that query dataset will map unambiguously a subset of their genes to a subset of the atlas genes.

- *Feature* genes. These are genes that serve as features for describing atlas metacell distributions. They represent the pool of features that can be used by the projection algorithm for positively linking query metacells with the atlas.
- *Noisy* genes. These are genes which are considered to be highly variable (e.g., bursty) across atlas cells, such that their incompatible expression in a query metacell may not be used as a negative indication for query/atlas match (e.g., TCR genes).
- *Lateral* genes. These are genes that are considered to be biologically irrelevant to the atlas, such that their incompatible expression in a query metacell may not be used as a negative indication for query/atlas match (e.g., cell cycle genes).
- *Excluded genes.* These genes were removed during the construction of the atlas and hence do not appear in it at all. In particular, these genes were not used when normalizing other genes to compute and compare metacell expression profile (e.g., mitochondrial genes).

B. A set of *metacells*, each defining a distribution of UMIs counts over the set of all (non-excluded) genes. Atlas metacells are usually linked in a graph structure defining metacell adjacencies/similarities, but the atlas projection functionality we describe below is not using these.

C. Metacell *type* annotations, a designated type name annotation for each metacell.

D. Per-type *essential genes*. For each type, a short list of genes can be defined as essential for that type; that is, every query metacell of that type is expected to have an atlas-compatible expression of these genes. For example, CD3D is an essential T-cell gene. Note there is no threshold specified, and that this association is one-way, that is, it is possible for metacells of other types to also express this gene.

### Representing a query datasets

A single cell data set can be used to query an atlas with MCProj by first transforming single cells into a metacell model using metacell-2 or any approach for grouping together similar single-cell profiles and inferring their expression distribution. This is an essential step of the process, as the projection algorithm requires robust gene expression level profiles for both the query and the atlas, and single-cell profiles are not robust enough to generate reliable results.

The set of genes in the query dataset should overlap extensively with the atlas features. Some standard gene name translation can be applied if the query is derived over a different gene name space from the atlas. MCproj is however compatible with query gene sets that are only partially matching the atlas genes (such as in different versions of genome assembly or annotations), in which case only genes in the intersection will be considered for computing query/atlas match.

### Projection algorithm overview

The overall projection algorithm consists of the following steps:

1. Initialization: Find the set of genes common to the query and atlas metacells. All computations will be done on this common subset. Pick the initial set of genes to use out of this subset.
2. Seeding and mixture modeling: Ignoring the type annotations of the atlas, compute a preliminary projection for each of the query metacells.
3. Multiplicative gene correction: Adding a multiplicative correction to genes that are highly correlated between the query and atlas, but still show a magnitude change.
4. Range-based filtering: Filter out genes whose expression range in the query and the atlas are significantly different.
5. Filtering genes within atlas types: Refine the preliminary projection using the type annotations of the atlas. This includes computing QC measures for the refined projection.
6. Composite projection: For each query metacell, which we failed to project to a single region of the atlas, attempting to project it to a composition of two atlas regions. This implies we try to model it as a metacell combining the two distinct atlas state regions, due to doublet cells grouped together to form a metacell, or to due to a mixture of cells of different states that could not be split into two when analyzing the query.

Below we describe these steps in more detail. When computing fold factors, we use a regularization factor of 1e−5. We also ignore fold factors for genes where the sum of

UMIs in the two profiles is not at least 40. All the listed parameter values (e.g., the normalization factor and the UMIs threshold) can be modified by the user.

**Projection initialization**

We define the set of common query and atlas genes as $G$. From the UMI $U$ matrix over gene and query metacells, we compute the query gene expression matrix:

$$e_{gi}^Q = log2(\epsilon + \frac{u_{gi}^Q}{N_i^Q})$$

where $N_i^Q = \sum_g u_{gi}^Q$ represents the total number of UMI in query metacell $i$ over the common $G$. Similarly, we define the atlas expression matrix:

$$e_{gj}^A = log2\left(\epsilon + \frac{u_{gj}^A}{N_j^A}\right).$$

Note that here again the denominator $N_i^A = \sum_g u_{gi}^A$ is computed using the same set of common genes $G$. The regularization parameter $\epsilon$ is set to $10^{-5}$ by default.

**Seeding and mixture modeling**

Let the correlation matrix between the query and the atlas be defined as:

$$c_{ij} = cor(e_{gi}^Q, e_{gj}^A)$$

Mixture modeling is approached independently for each query metacell. For each such metacell $i$ we identify the atlas metacell $anchor_i = argmax_j(c_{ij})$. We then extend the anchor atlas metacell into a set of candidates $C_i$ using distance between the anchor and atlas metacells $d_{ij} = \max\left(abs\left(e_{g,anchor_i}^A - e_{gj}^A\right)\right)$, selecting the closest 10 atlas metacells, and adding to them the closest 40 additional metacells with $d_{ij} < 2$, if such metacells exist.

We can compute the optimal representation of the query as a weighted average of these candidates:

$$e_{gi}^P = \sum_{k \in C_i} w_{ik} e_{gk}^Q$$

by minimizing the L2 distance between the two vectors $\|e_{gi}^Q - e_{gi}^P\|_2$, subject to:

$\sum_k w_{ik} = 1, w_{ik} \geq 0$ using a convex solver (specifically, Python's cvxpy solver). No regularization term is needed given the non-negativity constraint, but we nevertheless clean up weights smaller than 1e−5 and renormalize.

The optimal representations of all query metacell are defining together a matrix $E^P$ that we can now compare to the query expression matrix $E^Q$.

**Multiplicative gene correction**

If gene correction is needed (as determined by a user parameter), we compute for each gene the correlation between its query and projection expression over all metacells

$fit_g = cor(e^Q_{ig}, e^P_{ig})$ and its overall expression fold factor $cf_g = \sum_i e^P_{ig}/\sum_i e^Q_{ig}$. Genes with $fit_g > 0.8$ and a high degree of overall fold change $(abs(log(cf_g)) > log(1.15))$ are corrected by setting $e'^Q_{ig} = cf_g e^Q_{ig}$. Given corrected query data, we can reiterate the seeding and mixture phase. We repeat this procedure at most three times (or less if no genes are corrected).

### Range-based filtering

For each projected gene, we compute the range of its (possibly corrected) query expressions $[low^Q_g = quantile(e^Q_{ig}, 2\%)\ldots high^Q_g = quantile(e^Q_{ig}, 98\%)]$ and similarly its projected range $[low^P_g = quantile(e^P_{ig}, 2\%)\ldots high^P_g = quantile(e^P_{ig}, 98\%)]$. If the shared expression range $[low^S_g = max(low^P_g, low^Q_g)\ldots high^S_g = min(high^P_g, high^Q_g)]$ is less than 50% of the query range, then we filter out the gene as being too different between the query and the atlas.

### Filtering genes within atlas types

After deriving the initial projection weights we can compute for each metacell and gene the query/projection log fold change as $lfold_{gi} = log2((\epsilon + e^Q_{gi})/(\epsilon + e^P_{gi}))$. We identify genes that are significantly skewed in the query compared to their projection as $\delta_{gi} = abs(lfold_{gi}) > 3$.

We next aim at filtering genes whose distribution in the query is completely skewed compared to the atlas. This is however approached within each atlas type independently to maximize sensitivity. We note that genes that are tagged as skewed for a given type at this stage are still available for downstream analysis by the user. This should account for both queries in which a gene is perturbed as part of the experimental design (i.e., knockout), or alternatively to data that involves significant ambient noise in some types. To that end, we used the computed projection weights to determine for each query metacell the atlas annotation term with maximal weight ($type(i)$), thereby grouping query metacells into atlas types. We then compute for each type and gene the fraction of metacells that are highly skewed $skew_{gt} = mean_{type(i)=t}(\delta_{gi})$, and define genes for which $skew_{gt} > 0.5$ as $G^{skew}_t$.

Having computed the skewed gene sets for all types, we repeat the seeding and mixture step of the algorithm, while filtering for each query metacell the genes from its respective skewed set. We note that this may result in changing the assigned type for some query metacells. In that case, we repeat the entire iteration until convergence (but at most three times), searching for additional skewed genes in each type.

The endpoint of this step are updated projection weights $w_{ij}$, and a combined set of skewed genes for each type $G^{skew}_t$.

### Composite projection

We update the thresholded skew matrix $\delta_{gi}$ by setting all values for type-skewed genes ($G^{skew}_t$) to 0 for metacells assigned with the respective type ($type(i) = t$). Metacells for which $\sum_g \delta_{gi} > 3$ are defined as unassigned (the threshold is user adjustable). We also define as unassigned metacells for which 25% of the essential genes of its assigned are

skewed (recall that the atlas definition may include a list of essential genes for each type annotation term).

Unassigned metacells may represent behaviors that are simply not represented in the atlas, but some of them may instead be an outcome of composing expression distributions from two distinct regions in the atlas. Such composite behaviors are not initially allowed by MCProj – since it is assumed assigned query metacells should map to a unique atlas state. Composite behaviors may originate from clusters of doublets in the query or can represent groups of single cells that could not be split by metacells algorithm (due to sparsity, low state frequencies or noise).

To assist users in distinguishing composite behaviors from true novel states in the query, we model unassigned metacells as more flexible mixtures of atlas metacells from two distinct atlas regions. This is done by (a) subtracting $e_{gi}^P$ from the query and atlas expression distribution, (b) finding a second set of candidate atlas metacells using correlation between the residual expression distributions, (c) fitting the query as a weighted average of the union of first and second set of candidate metacells (this time working with the original, rather than the residual values).

Unassigned metacells for which this strategy derives faithful projection (not more than 3 and up to 25% essential skewed genes), we declare the query metacell as a composite and assign it a combination of two types, based on its weights in the 1st (primary) and 2nd (secondary) atlas regions. Otherwise, we use the original representation of the query metacells using atlas metacells from a single region.

### Projections QC

The core result of the algorithm is the matrix of weights $w_{ij}$ used to best represent each query metacell as a weighted average of several atlas metacells, together with the (corrected) query UMIs ($E'^Q$) and the projected UMIs ($E^P$).

It is however necessary to determine whether this representation is acceptable for each query metacell. To assist in this decision, the algorithm provides a set of per-metacell and per-gene annotations, described below:

#### *Per metacell annotations*

1. The *projected type* assigned to the query metacell by the projection. This is the best label we can assign to it, if we are forced to, ignoring any QC indicators.
2. The *composite type* assigned to the query metacell, or the empty string if there is none (most of the cases).
3. A mask of the *fitted genes* of the metacell. This mask is the same for all metacells of the same projected type and no composite type (the normal case). For composite query metacells, this mask is the intersection of the masks of the projected and composite types.
4. A *misfit genes count* of all the fitted genes whose fold factor between the query and its projection is above a threshold of 8x ($sum_g(\delta_{ig})$).
5. A *misfit essential genes fraction* containing the fraction of essential genes (of both the projected and composite type, if any) whose fold factor between the query and its projection is above 8x.

6. A mask of the *similar* (successfully projected on the atlas) metacells, which have a low number of misfit genes and misfit essential genes, and which fitted at least 1/3 of the feature genes of the query.

### *Per gene annotations*

1. A mask of *atlas genes* of the genes common to the query and the atlas.
2. Masks of *atlas feature genes*, which were used to initialize the fitted genes masks.
3. A mask of *correlated genes* that had high correlation between the query and the projection (and therefore were candidates for linear scale correction).
4. The *correction factor* applied to the genes, if highly correlated and requiring a correction (1 if not corrected).
5. A mask of *fitted genes per type* for convenience (used as the mask of fitted gene per metacell).

The analyst is expected to use the misfit QC measures to determine when to accept the projected type, as well as comparing the UMIs profile of the (corrected) query to the UMIs profile of its projection on the atlas. In addition, since the misfit QC measures depend on the fold factor threshold used during the projection, the analyst may choose to re-compute them using a different threshold for making this determination, based on the above UMIs profiles.

## Supplementary Information

---

**Additional file 1: Fig S1.** Validation: leave-one-batch-out projections. Similar to Fig. 2A. Counting cells classified according to their reference annotated type and the type assigned to their query metacell by scARCHES. **Fig S2.** Validation: leave-one-batch-out projections. Similar to Fig. 2A. Counting cells classified according to their reference annotated type and the type assigned to their query metacell by Seurat. **Fig S3.** Validation: leave-one-batch-out projections. Similar to Fig. 2A. Counting cells classified according to their reference annotated type and the type assigned to their query metacell by scMAP. **Fig S4.** Validation: leave-one-type-out projections. Similar to Fig. 3A. Counting cells classified according to their reference annotated type and the type assigned to their query metacell by scARCHES. **Fig S5.** Validation: leave-one-type-out projections. Similar to Fig. 3A. Counting cells classified according to their reference annotated type and the type assigned to their query metacell by Seurat. **Fig S6.** Validation: leave-one-type-out projections. Similar to Fig. 3A. Counting cells classified according to their reference annotated type and the type assigned to their query metacell by scMAP. **Fig S7.** The fraction of query cells for which a synthetic technology difference was applied (and was not corrected), which were either not successfully projected, or which were assigned a type different than the expected type, for each of the methods we compared, very close to Fig. 2B.

**Additional file 2.** Review History.

---

### Acknowledgements
We thank members of the Tanay group for their comments and help with preparing this manuscript.

### Peer review information

### Review history
The review history is available as Additional file 2.

### Authors' contributions
OBK and AT designed the MCProj algorithm and the study. OBK implemented the algorithm and the simulations. AL implemented downstream analysis code and visualization tools. AB, OR, and DB worked on testing and improving MCProj behavior with real data. OBK and AT wrote the paper with input from all authors.

## Funding
Research was supported by the European Research Council (cells2Tissues, 101054948), the Israel science foundation IPMP and Breakthrough programs, and the EU BRAINTIME 874606 project.

## Availability of data and materials
Code and documentation of our new tools are available under the MIT license at https://github.com/tanaylab/metac ells [20] and Zenodo [21]. Vignettes demonstrating the use of the metacells package and employing the MCProj method are available at https://github.com/tanaylab/metacells-vignettes [22] and Zenodo [23]. An interactive view of the results listed above (which were computed using version v0.9.0 of the package), using the MCView tool avail-able at https://github.com/tanaylab/MCView [24] and Zenodo [25], is available at https://apps.tanaylab.com/MCV/mcproj/paper/index.html [26].

# Declarations

## Ethics approval and consent to participate
No ethical approval was required for this study.

## Competing interests
The authors declare that they have no competing interests.

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

## 
