## [**Additional file 2.** Review History. · Genome Biology]

Review History

First round of review

Reviewer 1

Were you able to assess all statistics in the manuscript, including the appropriateness of statistical tests used? Yes.

Were you able to directly test the methods? No.

Comments to author:

This manuscript describes a novel approach for characterizing scRNA-seq data with reference to existing data, called MCProj. The analysis revolves around the use of "metacells" - a concept the authors have previously established to define a set of cell states within single-cell RNA-seq data via similarity analysis and pseudo-bulk aggregation. Here, the authors represent the existing reference data as a manifold of metacell gene expression distributions, then infer metacells in the query dataset as mixtures of states in the reference, correcting for global biases in a gene-specific manner. In addition to defining common states, this approach flags cells and gene expression patterns that are not observed in the reference, which has emerged as an issue with batch correction ("integration") and label transfer methods.

Overall, I found this method to be clever, straightforward, and likely to be of broad utility. The manuscript is well written and will be of interest to a broad audience of biologists looking to compare their data to existing atlases. The software is easy to install and use. To make this method even more accessible, I strongly recommend creating a short vignette demonstrating how to use MCProj, that takes users from making a reference atlas through to querying and interpreting outputs. I have also included some minor points that could be added as discussion points, but are not necessary revisions.

Major issues

- Multiplicative correction simulations (4EF) - why are baselines different between with and without correction, even when the simulated effect strength=0? This seems like an error. If this cannot be explained, I would recommend removing this simulation as it does not add much to the paper or to the interpretability of the method.^[L]_[SEP]

- I was able to install metacells and run through the metacells vignette. It would be very helpful for users if the authors could make an MCProj-specific vignette.

Minor issues/discussion questions

- Regarding MCProj's multiplicative correction: This seems to work well for 10x v2 vs. v3, and I can imagine it would work well for mismatched read-depths between query/atlas datasets, but how likely is this to generalize? Have you looked at e.g., single nucleus vs. single-cell RNA-seq? Since comparing data between nuclei and cells is likely to be a fairly common use case, it may warrant additional discussion or an example.^[L]_[SEP]

- Panel 4D is not very informative - In this plot I can see that the correlations increase after correction, but I don't see any relationship between the y-axis and the x-axis. It's helpful to have a sense of the correlation distributions before and after correction, as shown in 4B. Otherwise this plot makes me curious whether there's any relationship between the size of the correction and the corrected vs. uncorrected correlation, e.g., plot corrected-uncorrected residual vs. $\text{abs}(\log_2 \text{ gene correction})$, maybe colored by corrected cor >0.8 - do larger corrections yield larger improvements in correlations, or is there no relationship here? Do the authors think there is anything to be learned about genes or technical errors from similarity in the correction factors? This is just a question for discussion, not a necessary revision.^[1]_[SEP]

- Benchmarking tests are nicely designed (donor hold-out and cell-state hold-out). Can you comment on how this is expected to perform with disease-associated states? (e.g., Dann et al 2022 bioRxiv that shows transfer learning approaches do better when there is matched normal data in the query). My assumption is that this is simply a special case of cell-state hold out but it would be helpful to hear the authors' thoughts on this specifically (or better yet - to include this as another example).

- Interactive examples could not be accessed with the provided link (<http://tanaylab.weizmann.ac.il/mcproj>) but I did find them through the MCView github <https://github.com/tanaylab/MCView> e.g., <https://tanaylab.weizmann.ac.il/MCV/PBMC/>. Please fix broken links before publication.

Reviewer 2

Were you able to assess all statistics in the manuscript, including the appropriateness of statistical tests used? Yes: To my knowledge, I found the methods very sound, and well explained in the methods section.

Were you able to directly test the methods? No.

Comments to author:

The manuscript by Ben-Kiki and colleagues explores how a meta-cell based strategy (MCProj) can improve the projection of cells from query single-cell data onto a reference single-cell atlas. They show that MCProj can accurately project query data after the meta-cells in the query data have been identified and then projected onto the reference. It can handle batch effects and if the reference atlas lacks a specific cell-type that is present in the query, a large fraction of cells (of the query meta cell) are left unassigned - indicative of lacking a reference counterpart. It can also handle gene-level, systematic biases across technologies by introducing a scaling procedure.

I think the overall strategy and rationale of this paper is very sound - great work. Since individual cells can be seen as sampled outcomes from the cell state, the ability to perform comparisons on the level of cell state descriptions, and not individual cells, is definitely a good approach. The manuscript provides compelling evidence that MCProj can improve accuracy of projections, compared to other commonly used strategies, and better handle cell type that is lacking in the reference.

Main comments

1. Information on genes and cells identified in analyses of HCA BM data.

It would be great to report the number of genes classified while processing the HCA BM data, e.g. how many were identified as "noisy", "lateral", "excluded" or "feature" genes etc.

2. Query cells not part of meta-cells

Cells deemed noisy or novel and not associated with any meta-cell in the query, are they removed from the analysis in MCProj or are they present in the unassigned category. In comparisons with other methods, the exact same cells are always kept and made statistics from. Just making this explicit would be good.

3. Explanations of figures

The manuscript would benefit from more explicit descriptions of certain figures, in particular Figure 2a/Figure3a are complex and two more sentences in the legend would definitely help, now readers have to spend some time figuring out the various figure items. For example, the colorbars on the far right have no label (presumably, number of cells). That columns contains the results of each leave-on-out experiment, on each cell-type, respectively, the two fractions of cell barplots can be explicitly described, etc...

Other figures are easier to follow.

4. Correct for larger technology-associated biases

The authors compared 10x v2 and v3 and found that technology-associated systematic biases could to some extent be corrected for within MCProj. Since v2 and v3 in 10x are still more similar that a comparison that included other data, it would be interesting to see more diverse datasets compared.

Typos encountered:

Line 13, page of Fig 1A-C referencing: "features, It".

Line 35, paragraph with Fig2B reference: double word: "using".

Line 46, Sentence above Fig3D reference: pumicated -> punctuated?

Dear Editors,

We are delighted to submit the revision for our paper "*Metacell projection for interpretable and quantitative use of transcriptional atlases*". Following the highly supportive feedback from the reviewers, we have improved the paper to meet the remaining concerns:

- We have added thorough vignettes describing the use of our new tool;
- Added an interactive website allowing exploration of our results in <https://apps.tanaylab.com/MCV/mcproj/paper/index.html> and in <https://apps.tanaylab.com/MCV/mcproj/review/index.html>
- Added more analysis of projections between different technologies to our response below, highlighting MARS-seq over 10X and snRNA-seq over scRNA-seq use cases.
- In the revised results we have refined our gene filtering policy to better accommodate queries that are using markedly different technologies, leading also to an overall improved cross validation.

We apologize for the delay in resubmitting. We worked hard to improve the code itself and its robustness. We are grateful for your help with handling the paper and are looking forward to seeing it in press soon.

Sincerely,

Oren Ben Kiki and Amos Tanay

Reviewer #1: This manuscript describes a novel approach for characterizing scRNA-seq data with reference to existing data, called MCProj. The analysis revolves around the use of "metacells" - a concept the authors have previously established to define a set of cell states within single-cell RNA-seq data via similarity analysis and pseudo-bulk aggregation. Here, the authors represent the existing reference data as a manifold of metacell gene expression distributions, then infer metacells in the query dataset as mixtures of states in the reference, correcting for global biases in a gene-specific manner. In addition to defining common states, this approach flags cells and gene expression patterns that are not observed in the reference, which has emerged as an issue with batch correction ("integration") and label transfer methods.

Overall, I found this method to be clever, straightforward, and likely to be of broad utility. The manuscript is well written and will be of interest to a broad audience of biologists looking to compare their data to existing atlases. The software is easy to install and use. To make this method even more accessible, I strongly recommend creating a short vignette demonstrating how to use MCProj, that takes users from making a reference atlas through to querying and interpreting outputs. I have also included some minor points that could be added as discussion points, but are not necessary revisions.

We thank the reviewer for the support and for the helpful comments. See detailed responses below.

Major issues

- Multiplicative correction simulations (4EF) - why are baselines different between with and without correction, even when the simulated effect strength=0? This seems like an error. If

this cannot be explained, I would recommend removing this simulation as it does not add much to the paper or to the interpretability of the method.

Thank for pointing this out. The reason for the baseline differences is a small degree of overfitting that is being introduced whenever the multiplicative correction is in effect. We control overfitting by using a very restricted form of correction and by applying correction only for genes with strong support for it, but some remaining effect can still be observed. This is also why it is important to include this simulation – we have clarified this explicitly in the revised text.

- I was able to install metacells and run through the metacells vignette. It would be very helpful for users if the authors could make an MCPProj-specific vignette.

Point well taken. We have added detailed vignettes here:
<https://github.com/tanaylab/metacells-vignettes>
And the MCView applications can be viewed at:
<https://tanaylab.weizmann.ac.il/mc2.vignettes>

Minor issues/discussion questions

- Regarding MCPProj's multiplicative correction: This seems to work well for 10x v2 vs. v3, and I can imagine it would work well for mismatched read-depths between query/atlas datasets, but how likely is this to generalize? Have you looked at e.g., single nucleus vs. single-cell RNA-seq? Since comparing data between nuclei and cells is likely to be a fairly common use case, it may warrant additional discussion or an example.

We first note that in the revised paper we refined the MCPProj gene filtering process in order to distinguish genes that can be used for matching query and atlas from those that cannot be used (given clear difference in their distributions which cannot be corrected via a multiplicative factor). This has a positive effect on our cross-validation analysis (see updated Figure 1 and Figure 2) and on the application of MCPProj to other manifolds not described here. We have added more discussion around the important point of technology correction, and in particular to the question of mapping snRNA-seq on scRNA-seq data. To demonstrate it here, we mapped mouse gastrulation snRNA-seq on our scRNA-seq gastrulation atlas.

Our recommended minimal fraction of fitted gene out of genes that show variable expression in the query is 33%, and the multiome projection is associated with only 14% (17% with correction) which suggests multiome projection on an scRNA-seq atlas will initially be flagged by MCPProj as “unassigned”. To examine more closely the performance of MCPProj in such scenario we relaxed the threshold and derived a highly consistent result (<https://apps.tanaylab.com/MCV/mcproj/review/nucleus-uncorrected>, <https://apps.tanaylab.com/MCV/mcproj/review/nucleus-corrected> and Fig p2p_1 below). The derived projection leaves out a large number of genes that are non-linearly modulated between the query and projection, possibly due to post-transcriptional control. We suggest that in depth analysis of such controls is out of the scope of the current work and will be best approached through follow up studies by others or us.

- Panel 4D is not very informative - In this plot I can see that the correlations increase after correction, but I don't see any relationship between the y-axis and the x-axis. It's helpful to have a sense of the correlation distributions before and after correction, as shown in 4B. Otherwise this plot makes me curious whether there's any relationship between the size of the correction and the corrected vs. uncorrected correlation, e.g., plot corrected-uncorrected residual vs. $\text{abs}(\log_2 \text{ gene correction})$, maybe colored by corrected cor >0.8 - do larger

corrections yield larger improvements in correlations, or is there no relationship here? Do the authors think there is anything to be learned about genes or technical errors from similarity in the correction factors? This is just a question for discussion, not a necessary revision.

We apologize for the confusion here. Figure 4D is mostly intended for explaining which genes are selected for correction – in the context of all genes. The impact of the correction can be read from Fig 4E and 4F (not for individual genes, but for cell states). We have clarified this in the text.

- Benchmarking tests are nicely designed (donor hold-out and cell-state hold-out). Can you comment on how this is expected to perform with disease-associated states? (e.g., Dann et al 2022 bioRxiv that shows transfer learning approaches do better when there is matched normal data in the query). My assumption is that this is simply a special case of cell-state hold out but it would be helpful to hear the authors' thoughts on this specifically (or better yet - to include this as another example).

We are using MCPProj extensively to project patient query on an atlas of blood HSPC cells, for example. We thought that adding a disease state to the paper will lose coherence since it will require in depth discussion of patient specific behaviors and distinguishing several types of such variation in a way that use projection as a basis but add to it also tests of patient state composition.

- Interactive examples could not be accessed with the provided link (<http://tanaylab.weizmann.ac.il/mcproj>) but I did find them through the MCVIEW github <https://github.com/tanaylab/MCVIEW> e.g., <https://tanaylab.weizmann.ac.il/MCV/PBMC>. Please fix broken links before publication.

Thank you. Links are now alive at <https://apps.tanaylab.com/MCV/mcproj/paper/index.html>

Reviewer #2: The manuscript by Ben-Kiki and colleagues explores how a meta-cell based strategy (MCPProj) can improve the projection of cells from query single-cell data onto a reference single-cell atlas. They show that MCPProj can accurately project query data after the meta-cells in the query data have been identified and then projected onto the reference. It can handle batch effects and if the reference atlas lacks a specific cell-type that is present in the query, a large fraction of cells (of the query meta cell) are left unassigned - indicative of lacking a reference counterpart. It can also handle gene-level, systematic biases across technologies by introducing a scaling procedure.

I think the overall strategy and rationale of this paper is very sound - great work. Since individual cells can be seen as sampled outcomes from the cell state, the ability to perform comparisons on the level of cell state descriptions, and not individual cells, is definitely a good approach. The manuscript provides compelling evidence that MCPProj can improve accuracy of projections, compared to other commonly used strategies, and better handle cell type that is lacking in the reference.

We thank the reviewer for the support and helpful comments. Detailed answers are provided below.

Main comments

1. Information on genes and cells identified in analyses of HCA BM data.
It would be great to report the number of genes classified while processing the HCA BM

data, e.g. how many were identified as "noisy", "lateral", "excluded" or "feature" genes etc.

We added this information to the text.

2. Query cells not part of meta-cells

Cells deemed noisy or novel and not associated with any meta-cell in the query, are they removed from the analysis in MCProj or are they present in the unassigned category. In comparisons with other methods, the exact same cells are always kept and made statistics from. Just making this explicit would be good.

Thank you for this comment. MCProj is initially running metacell analysis on the query, which is critical for defining quantitative states (and for their projection) and for filtering problematic, noisy and outlier cells from further analysis. This is an integral part of the pipeline – and it is indeed very important to explain it more clearly as now done in the revision. We note that the atlas is also going through a similar process, since the goal is to compare distributions of gene expressions while minimizing confounders due to outlier cells.

3. Explanations of figures

The manuscript would benefit from more explicit descriptions of certain figures, in particular Figure 2a/Figure3a are complex and two more sentences in the legend would definitely help, now readers have to spend some time figuring out the various figure items. For example, the colorbars on the far right have no label (presumably, number of cells). That columns contains the results of each leave-on-out experiment, on each cell-type, respectively, the two fractions of cell barplots can be explicitly described, etc...

Other figures are easier to follow.

Thank you for point this out. We have added explanation to the legends and added annotations.

4. Correct for larger technology-associated biases

The authors compared 10x v2 and v3 and found that technology-associated systematic biases could to some extent be corrected for within MCProj. Since v2 and v3 in 10x are still more similar that a comparison that included other data, it would be interesting to see more diverse datasets compared.

We are routinely using MCProj to project MARS-seq data on 10X and vice versa. See **Fig p2p_2** and the interactive maps in

<https://apps.tanaylab.com/MCV/mcproj/review/gotg-uncorrected>
and <https://apps.tanaylab.com/MCV/mcproj/review/gotg-corrected>

We agree that correcting technologies that are significantly different in some major non-linear aspect due to regulation must be approached differently. This can be the case for snRNA-seq vs. scRNA-seq, or for techniques using 5' utr sequencing, whole body transcriptomics etc. In such cases the MCProj approach is to restrict the projection fit to genes that are reasonably distributed across the query and atlas (see revised method), and to require that a large fraction (at least 1/3 by default) of the query genes are fitted successfully. All other genes can be examined in order to detect the technological or biological effects that diversify their expression.

We applied this approach to project a mouse gastrulation multiome data over RNA-seq (MARS-seq based) atlas (see **p2p_1** and

<https://apps.tanaylab.com/MCV/mcproj/review/nucleus-uncorrected> and <https://apps.tanaylab.com/MCV/mcproj/review/nucleus-corrected>). This provided very consistent cell type/state matching and high degree of quantitative correlation on the fitted genes. We note that in order to project this data we had to relax the threshold for

minimal percentage of fitted feature genes to 1/8 (default 1/3). In normal use cases, we suggest users will be prompted with an “unassigned” result for such scenario, and would be able to follow up with specific analysis of the differentially profiled or regulated genes.

Typos encountered:

Line 13, page of Fig 1A-C referencing: "features, It".

Line 35, paragraph with Fig2B reference: double word: "using".

Line 46, Sentence above Fig3D reference: pumicated -> punctuated?

Fig p2p_1

The distribution of R^2 of fitted genes when projecting nucleus-only (multiome) data set on the bone marrow data, with and without applying the technology correction. Note that MCPProj used only 14% and 17% of the query genes for fitting in this use case.

Fig p2p_2

The distribution of R^2 of fitted genes when projecting 10X data set on a MARS atlas, with and without applying the technology correction. Note that MCPProj used 34% and 38% of the query genes for fitting in this use case.

Second round of review

Reviewer 2

I think the authors have done a great job of extending their analyses to additional data types (MARS-seq, nuclei etc) that demonstrates that general improvements beyond 10x versions. Moreover, I think the additional information provided (figures) and text explanations, have improved the manuscript. I think the manuscript is ready to be accepted.